





# Offshore wind farm cluster wakes as observed by a long-range scanning wind lidar

Beatriz Cañadillas[1,2], Maximilian Beckenbauer[1], Juan J. Trujillo[2], Martin Dörenkämper[3],
Richard Foreman[2], Thomas Neumann[2], and Astrid Lampert[1]

[1]Institute of Flight Guidance, Technische Universität Braunschweig, Germany
[2]Renewables, UL International GmbH, Oldenburg, Germnany
[3]Fraunhofer Institute for Wind Energy Systems, Oldenburg, Germany

**Correspondence:** Beatriz Cañadillas (b.canadillas@tu-braunschweig.de, beatriz.canadillas@ul.com)

**Abstract.** To establish long-term flow measurements for the validation of wake models, a scanning Doppler wind lidar system was installed at the western edge of the wind farm Gode Wind 1 in the German Bight for a period of five months. The main goal was to detect the wakes from clusters for different wind directions and atmospheric stabilities. The lidar data are categorized

into five sectors based on the different upstream conditions. The influence of wakes and atmospheric stability are initially investigated with respect to airborne measurements collected within the lidar measurement period. Mesoscale simulations are used as a reference for the free wind flow. The percent wind speed difference downstream of the wind farm clusters and at the location of the scanning lidar measurements (1.5 km downstream the closest wind farm) can reach a maximum of about 30% for a mean wind speed of 10 m s$^{-1}$ depending on the wind direction and under stable atmospheric conditions. A good

agreement between mesoscale simulations (without any wind farm parameterization) and lidar measurements is found for undisturbed wind sectors and unstable and near-neutral atmospheric conditions. By taking into account the surrounding wind farms through a parameterization in the mesoscale simulations, the agreement of the model with the measurements is relatively good for unstable and near-neutral conditions, including sectors influenced by wind farm wakes. For stable conditions, however, the highest discrepancies between simulations and observations occur. Overall, the scanning lidar dataset can be used as a

validation tool for wake model validations.

## 1 Introduction

Offshore wind energy, i.e. the use of wind farms built offshore or on the continental shelf to harvest wind energy for electricity generation, is playing an important role in achieving a low-carbon future of economic prosperity. In 2020, 6.1 GW was commissioned worldwide. The total offshore wind capacity has now passed 35 GW, representing 4.8% of the total global cumulative

wind capacity. In particular, Germany represents a 22% contribution (7.8 GW) of the total installation (Lee and Zhao, 2021). In the North Sea, for instance, which can be transferred to other regions, the planned offshore areas for wind energy are becoming increasingly scarce. In order to contribute to the planned target of 30 GW by 2030 (long-term goal recently approved by the German government) and to make wind energy extraction economically profitable, wind farms need to be installed relatively



close to each other. While this may be beneficial in terms of infrastructure sharing, it may also be detrimental to the overall
energy extraction due to the influence of the wakes generated by the upstream wind farms.

Therefore, knowledge of the prevailing wind conditions is one of the crucial parts not only in the first phase of a potential
offshore wind farm to assess accurately the wind resource, but also during the operation phase of the wind farm. Although
numerical simulations and the detailed analysis of experiments in wind tunnels can provide good insight into the actual condi-
tions, high-quality in situ measurements in a real environment are essential. As the size of offshore wind farms increases and
they are grouped into larger arrays, also called clusters, wake effects take on greater importance, affecting not only the sur-
rounding wind conditions, but also reducing the efficiency of power generation for downstream wind farms. In the North Sea,
for example, the large size of wind farms and their proximity affect not only the performance of single downstream turbines
but also that of whole neighboring downstream farms (Cañadillas et al., 2020; Ahsbahs et al., 2020), which may reduce the
capacity factor by approximately 20% or more as suggested by Akhtar et al. (2021).

The effect of atmospheric stability on the extension of the wakes behind wind farms has been intensively studied in the
recent years through a number of analytical and experimental studies (Christiansen and Hasager, 2005; Emeis, 2010, 2018;
Djath et al., 2018; Ahsbahs et al., 2018; Nygaard and Newcombe, 2018; Cañadillas et al., 2020; Ahsbahs et al., 2020; Platis
et al., 2020), as well as numerical investigations (Patrick et al., 2014; Siedersleben et al., 2018b). For instance, Cañadillas et al.
(2020) analyzed data from a series of flights collected within the wakes at several downstream distances of two offshore wind
farm clusters located in the North Sea during different atmospheric stability conditions. They found that stable stratification
leads to significantly longer wakes with a slower wind speed recovery compared to unstable conditions. Their results reveal
that the average wake length (defined as the downstream distance where the wind speed has recovered to 95% of the freestream
wind speed) under stable conditions exceeds 50 km, while under neutral/unstable conditions, the wake length amounts to
around 15 km.

The analysis of wind farm cluster wake interaction is a complex task, as different interacting processes on multiple scales
have to be taken into account. On the one hand, these effects depend on climatological and seasonal changes, and on the other
hand, the wind farms extend over very large areas, experiencing natural spatial gradients with regard to the wind conditions.
Mesoscale models are capable of resolving effects that are relevant on these large scales using wind farm parameterizations
developed to account for the wind speed reduction and turbulence increase downstream of wind farms (Fitch et al., 2012;
Volker et al., 2015). A validation of simulations with airborne in situ data (Lampert et al., 2020) has been one of the aims of
the projects WIPAFF (Wind Park Far Field) and X-Wakes (Interaction of the wake of large offshore wind farms and wind farm
clusters with the marine atmospheric boundary layer) (Siedersleben et al., 2018b, a, 2020), and the airborne datasets have been
used as reference for the validation of simulations and parameterizations (Akhtar et al., 2021; Larsén and Fischereit, 2021).

Due to the high spatial and temporal resolution, long-range scanning Doppler wind lidars (also LiDAR, light detection and
ranging) have gained importance in the wind energy industry for a variety of applications, such as wind resource assessment
(Neumann et al., 2020), wind turbine and wind farm wake studies (Schneemann et al., 2020), and power performance testing
(Rettenmeier et al., 2014; Gómez Arranz and Courtney, 2021). Especially in the offshore sector, traditional masts are associated
with a high cost and long approval processes. In contrast, scanning wind lidars are cheaper, very flexible in terms of the scan





set-up and the installation (for instance, on a wind turbine transition piece), and easily accessible for system maintenance
during the maintenance routines of wind farms.

In the past, most studies, using scanning Doppler lidar, have been limited to investigations of the spatial wake characteristics
of isolated wind turbines (Wang and Barthelmie, 2015; Bastine et al., 2015; Bingöl et al., 2010; Käsler et al., 2010) or individual
wind farms (Smalikho et al., 2013; Aitken et al., 2014; Iungo and Porté-Agel, 2014; Herges et al., 2017; Krishnamurthy et al.,
2017; Zhan et al., 2020), such as the velocity deficit, the single wake extent (length and width) of a wake, and wake meandering
(Trujillo et al., 2010; Krishnamurthy et al., 2017) under various atmospheric conditions. More recently, lidars have also been
used to study the wind speed reduction upstream of a wind farm, the so-called blockage effect (Schneemann et al., 2021). Only
few studies have focused on the effects of cluster wind farm wakes on the wind speed (Schneemann et al., 2020), the value of
the scanning lidar measurements for validating wind farm parameterizations in mesoscale models (Goit et al., 2020) or simple
wake engineering model used for wind farm optimization and energy yield estimation (e.g. Brower and Robinson, 2012).

In this study, in order to determine the wake effects of interacting wind farms, data from different measurement locations
and methods are combined with the aim of obtaining a comprehensive picture of the wind situation in the region of an offshore
wind farm cluster.

Section 2 provides an overview of the locations and datasets used, including data of the scanning lidar, airborne measure-
ments and mesoscale simulations. Section 3.1 presents a direct comparison of the lidar data with high-resolution airborne data
in the vicinity of the measurement location; visualization of the aircraft measurements and WRF data give an example of the
spatial extent of wind farm cluster wakes. Section 3.2 reveals the influence of upstream wind farms by comparing the scanning
lidar data with mesoscale simulations with and without taking into account the wind farm wakes. In the conclusions (Section
4), the potential of the scanning wind lidar for validating wind farm parameterizations in numerical simulations is highlighted.

## 2   Site and methods

A field campaign using a scanning Doppler lidar was conducted at the western edge of the wind farm Gode Wind 1 in the
German Bight (see Figure 1) for a period of five months, from May to September 2020.

Additionally, data from the Dornier 128 D-IBUF research aircraft of the Technische Universität (TU) Braunschweig are used
for a particular period to extend the range of the available wind speed measurements both upstream, around, and within wind
farm cluster wakes. Due to the lack of a free wind reference (without any flow disturbance generated by the wind farms), high-
resolution mesoscale model data from the Weather Research and Forecasting model (without the wind farm parameterization,
hereafter WRF) were used. Additionally, the scanning wind lidar dataset is used to evaluate the WRF model outputs considering
wind farm wakes with the Fitch wind farm parameterization (Fitch et al., 2012), hereafter referred to as the WRF-WF set-up.

Starting from the location of the scanning lidar measurements, five different sectors were defined for the subdivision of the
measurement data into different wind direction regions. Figure 1 shows a map of the study area in the German Bight with the
defined wind direction sectors. The individual regions are labelled R1 to R5.

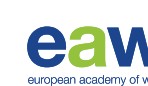
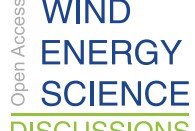

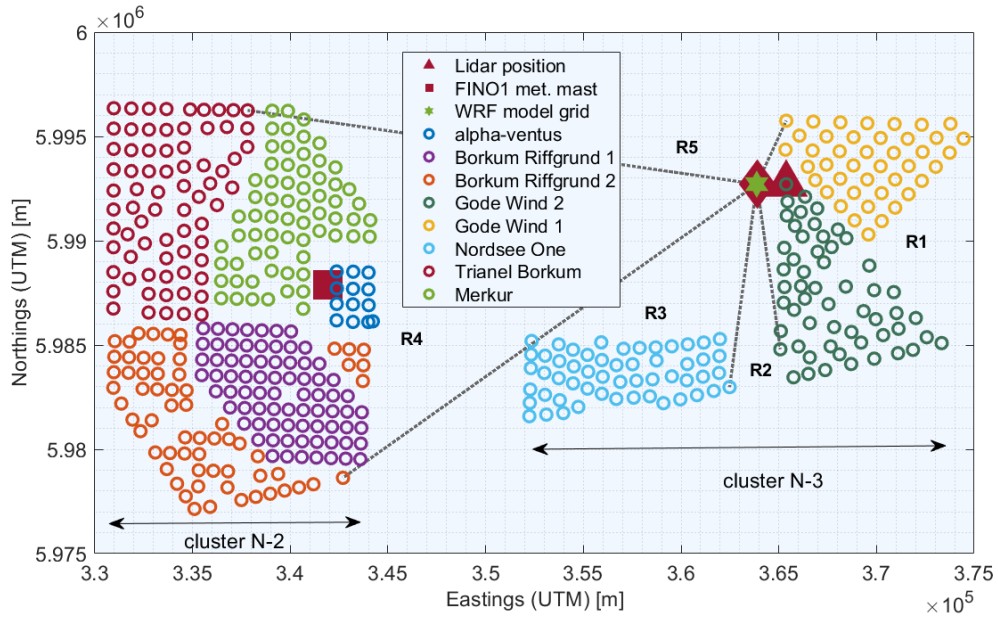

**Figure 1.** Location of the scanning lidar between the wind farm clusters showing the five sectors (from R1 to R5) into which the data are grouped for the analysis (dashed lines). The individual wind turbines are represented by circles, and each individual wind farm is shown in a different colours. The location of the meteorological (met.) mast FINO1 is also indicated (red square). The WRF model grid point investigated in this study is marked with a green star.

A detailed description of the measurement methods can be found in the following sections. For the classification into the five different regions according to the wind direction, the actual position of the profile measurements at a distance of 1.5 km west of the scanning lidar device was used. From here, the positions of the outermost wind turbines of each wind farm cluster were used to limit the area as presented in Table 1. The regions R2 (south of the lidar location) and R5 (north-west) are not influenced by upstream wind farms. Region R1 (east) is influenced by the wind farms Gode Wind 1 and 2. Region R3 is influenced by the wind farm Nordsee One, and region R4 by the wind farm cluster N-2 composed of the wind farms Trianel Borkum, Merkur, Alpha Ventus, and Borkum Riffgrund 1 and 2 (see Table 2 for a summary of the key characteristics of the wind farms surrounding the scanning lidar measurements).

After dividing the measurements into different regions based on wind direction, the lidar data have been further divided into subsets of atmospheric stability which is expected to strongly affect the wind speed downstream due to the presence of far-field wind farm wakes (Cañadillas et al., 2020).

In this study, we use the static atmospheric stability, which only takes into account buoyancy effects, and is characterized through the lapse rate ($\gamma$) based on the temperature gradient at two different altitudes (sea surface temperature (SST) and air temperature at the height of the transition piece (23.3 m) corrected for air pressure and density effects to obtain the virtual



**Table 1.** Ranges of the individual wind sectors (regions) and distances based on the measurement location of the scanner lidar system.

| Region | Sector boundaries [°] | Distance to lidar meas. point [km] |
|--------|----------------------|-----------------------------------|
| R1 | [24, 170] | 1.5 |
| R2 | [170, 186] | (free wind) |
| R3 | [186, 235] | 8 |
| R4 | [235, 277] | 20 |
| R5 | [277, 24] | (free wind) |

**Table 2.** Properties of the wind farms (as of May 2020) surrounding the scanning lidar measurement. The cluster name is defined by the German Federal Hydrographic Agency BSH, wind turbine (WT) type within the wind farms, WT rated power ($P_{rated}$), their rotor diameter ($D$), hub height ($h$) LAT (Lowest Astronomical Tide) and the number of wind turbines (number WT).

| Cluster | Wind farm | WT Type | $P_{rated}$ [MW] | $D$ [m] | $h$ [m] | Number WT |
|---------|-----------|---------|------------------|---------|---------|-----------|
| N-3 | Gode Wind 1,2 | Siemens | 6 | 154 | 110 | 55/42 |
| N-3 | Nordsee One | Senvion | 6.2 | 126 | 90 | 54 |
| N-2 | alpha-ventus | Senvion/Adwen | 5 | 126/116 | 92/90 | 6/6 |
| N-2 | Borkum Riffgrund 1,2 | Siemens / Vestas | 4 | 120/154 | 83 | 78/56 |
| N-2 | Trianel Windfarm Borkum 1,2 | Adwen/Senvion | 5, 6.3 | 116/164 | 87/111 | 40/32 |
| N-2 | Merkur | GE | 6 | 150 | 102 | 66 |

potential temperature ($\theta_v$) gradient,

$$\gamma = \frac{d\theta_v}{dz} \approx \frac{\Delta\theta_v}{\Delta z}, \tag{1}$$

with $z$ the measurement height. Negative values of the virtual potential temperature gradient $\gamma$, or lapse-rate, represent an
unstable stratification of the atmosphere, positive values represent a stable stratification and values around zero represent a neutral stratification. The stability classes were chosen as follows:

– $\gamma < -0.04$: unstable stratification

– $-0.04 \geq \gamma \leq 0.04$: near-neutral stratification

– $\gamma > 0.04$: stable stratification

We are aware of the low air temperature measurement height, ideally it would be optimal to measure the air temperature at hub height or above, but due to the lack of measurements and considering that the air temperature measurements at the nacelle are highly biased due to the rotor effect, we consider that our estimation is suitable as a first-order approximation and for the framework of this study.



## 2.1 Scanning wind lidar

Wind data were recorded with a long-range scanning Doppler wind lidar system of the type Streamline XR manufactured by Halo Photonics, UK (METEK-GmbH, 2021). The lidar system emits short laser pulses into the atmosphere and detects the radiation backscattered by aerosols through optical heterodyning. This makes it possible to determine both the intensity of the backscattered radiation and its Doppler shift in the line-of-sight (LOS) direction which is proportional to its radial wind speed, also called LOS wind speed.

The lidar system (see Figure 2) was installed on the transition piece (TP) of the northernmost wind turbine (K01) of the wind farm Gode Wind 1, at a height of approximately 23.3 m LAT (Lowest Astronomical Tide) and positioned on a metal support structure for a clear view over the railing to the west [160°, 20°]. In addition to the scanning lidar device, other sensors for collecting thermodynamic data (namely air temperature and humidity, pressure, precipitation and water surface temperature) were installed. The purpose of these measurements was to characterize the atmospheric stability regime with the method

previously described. The thermometer and hygrometer were mounted on a 50 cm long boom at a height of 22.5 m LAT and the barometer was located in the control cabinet 50 cm below. The infrared sensor for measuring the SST was located on the railing of the TP and consists of a pair of sensors, with one sensor pointed towards the sea surface and the other towards the sky which allows the temperature measurements to be corrected for the effects of background radiation (refer to Frühmann et al. (2018) for further details).

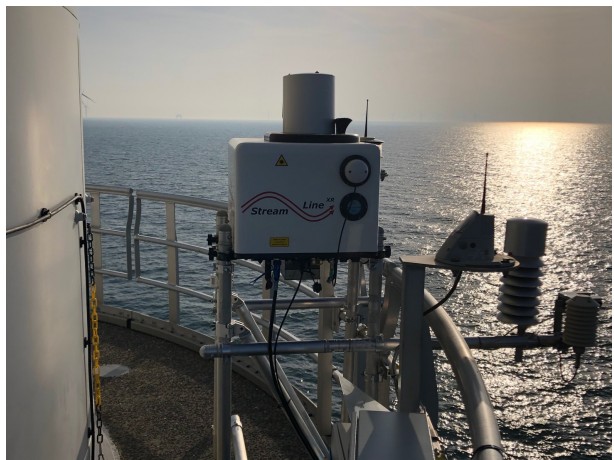

**Figure 2.** Long-range scanning lidar and additional measurement systems (on the right side) on the TP of one of the northernmost wind turbines (K01) at the wind farm Gode Wind 1.

The lidar system, with a maximum range of 10 km, was set up with a gate length of 120 m. The sampling rate of the backscattered signal of 50 MHz gives a spatial resolution of 3 m along the LOS. Furthermore, the accumulation rate can be reduced so that the highest beam sampling rate is 10 Hz. The laser beam is directed by a scanner with an arrangement of mirrors with two degrees of freedom, allowing scanning in all directions. The positioning of the scanner at the top of the lidar container box enables scanning of the sky above and a reduced area below the horizontal, without interference with itself.

The lidar performed plan position indicator (PPI) scans (at five elevations) with continuous scanner movement in two azimuthal sectors of 15° width upstream of the wind turbine K01. An overview of the lidar set-up is given in Table 3.



**Table 3.** Overview of scanning lidar set-up during the measurement campaign

| Parameter | Value |
|---|---|
| Target distance [m] | 1500 |
| Target heights [m] | 40, 80, 120, 160, 200 |
| Elevation angles [°] | 0.64, 2.17, 3.70, 5.22, 6.77 |
| Azimuth sectors [°] | 230°±7.5° & 300°±7.5° |
| Scan duration [s] | 75 |
| scan speed [° s$^{-1}$] | 3.17 |
| Accumulation time [s] | 0.6 |
| Range gate length [m] | 120 |

The set-up enabled the measurement of the wind profile in the vicinity of the wind farm Gode Wind 1 (approximately 1.5 km west of the wind turbine K01). To derive vertical profiles, we generate a so-called partial velocity azimuth display (VAD) plot

at several altitudes using a sinusoidal function fitted to radial velocity data (Werner, 2005), which is represented as a function of azimuth. Then the results are calculated in terms of the Cartesian velocity components ($u$, $v$, $w$) and finally the wind speed and direction are derived. The classical approach relies on four radial velocities measured at constant elevation and in four quadrants in the azimuth around the lidar. In our approach, we rely on several radial velocities measured continuously on a limited area in azimuth and constant elevation to avoid measurements influenced by the wake of the wind farms Gode Wind 1

and 2. The general sketch of the approach in Figure 3(a) shows how data are selected for the VAD plot near a target point and for several heights (b). After selecting data for an altitude, a check is made to see if the data meet a minimum carrier-to-noise ratio (CNR), and then a sine function is fitted using Random Sampling Consensus (RANSAC) (Fischler and Bolles, 1981). This last method is used to avoid the remaining outliers and to increase the robustness of the fitting procedure.

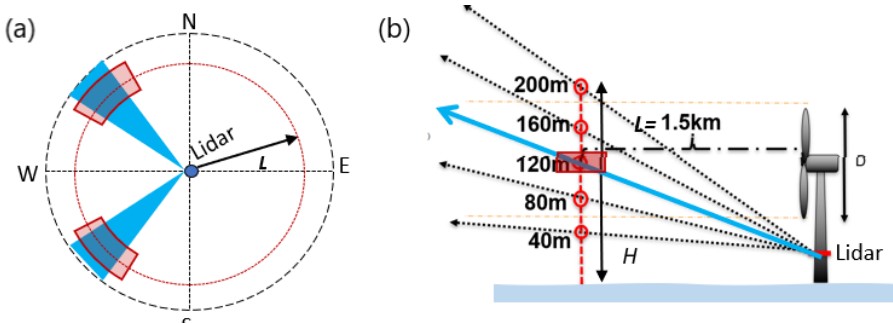

**Figure 3.** (a) Sketch of scanning for partial VAD. Shaded areas in blue represent areas where the wind is interrogated continuously. Areas in red represent the volumes where data for VAD are selected. (b) Sketch of the vertical profile of the wind speed and wind direction at the five target heights selected in this campaign, where $L$ is the distance from the lidar system to the measurement location.





As the lidar was positioned west of Gode Wind 1, the scanning was performed to the western side, targeting five heights
above LAT, namely at 40 m, 80 m, 120 m, 160 m and 200 m. In this free sector, we followed the scanning trajectory shown in
Figure 4(a) and with the scanner set-up shown in Table 3. In Figure 4(b), an example of the VAD results for the wind speed
and wind direction for 1500 m and a height of 120 m is shown.

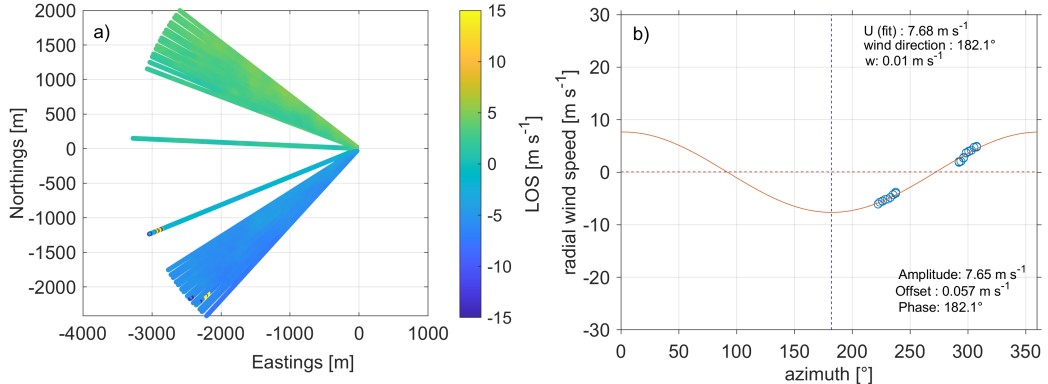

**Figure 4.** (a) Example of the top view of the radial wind speed ($u_r$) for a single full scan taking approximately 75 s on 29 August 2020.
Eastings and northings are given in meters and relative to the lidar position. (b) Example of VAD data selected from the scan in (a) for
a distance of 1500 m ± 150 m and a height of 120 m ± 5 m. The bottom legend shows results for the radial wind speed fit. The top legend
indicates results of the wind speed.

An important point to consider when measuring wind with a lidar system is the orientation of the system. Orientation errors
in the scanning lidar affect the exact position at which the wind is interrogated by the laser beam. Three angles are used to fully
define the orientation of the lidar in three-dimensional space, namely, bearing, tilt and roll.

The further the distance to the lidar, the larger the error in positioning is, due to errors in one of these angles. It is there-
fore necessary to determine these angles as at very high accuracy in order to reduce and properly quantify the positioning
uncertainty.

While the offset in the azimuthal direction between the geographic north and the lidar's north mark can be determined with
a compass, this is very inaccurate for the site installation because the turbine structure affects the magnetic field around the
lidar. A better option is to use the lidar itself and neighboring turbines of which their position is known ("hard targeting"). In
this study, we target turbines of the neighboring wind farm Nordsee One at distances between 8 km and 10 km and identified
them by their very high backscattered signal with an accuracy of at least $0.1°$.

In addition, the system is equipped with an internal inclinometer, which is used to quantify tilt and roll. However, the
manufacturer does not provide calibration information for this sensor. Furthermore, due to the high relevance of these angles,
it is desirable, if not mandatory, to perform an on-site assessment of mounting errors and inclinometer performance. For this
purpose, we apply the so-called Sea Surface Levelling (SSL) proposed by Rott et al. (2017, 2021) during the commissioning
of the lidar system at the offshore site. In this procedure, the sea surface is used as a reference to assess the orientation of the
lidar system relative to the horizontal plane. Mainly, the scanning lidar, which is installed several metres above the sea, is set to





perform a scan with constant downward elevation and constant azimuth velocity. In this set-up, the backscattered lidar signal describes the surface of a cone that extinguishes by absorption as it enters the water. The geometric analysis of the elliptical shape of the intersection of the cone with the water surface (see Figure 5) provides the tilt and roll angles of the cone axis and thus of the lidar itself.

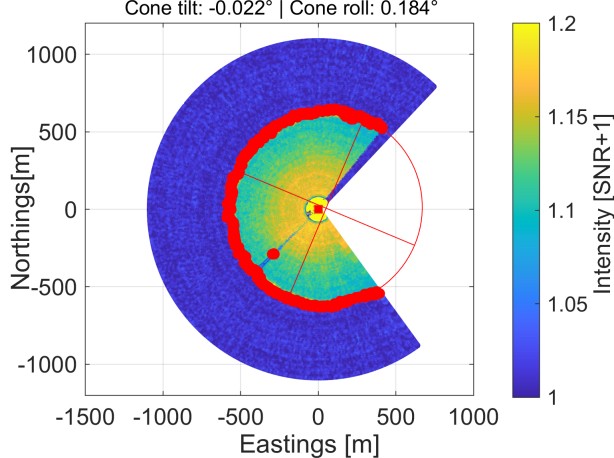

**Figure 5.** Example of backscattered signal intensity after SSL scanning. Red dots show the estimated water entrance. The red ellipse and its axes reveal a misalignment of the sensor. The blind area to the east is due to the turbine tower.

We adopt the results of the SSL method as a reference because they show the misalignment of lidar, scanner and support

structure combined in a direct way. For this reason, the misalignment results from the SSL can be used in trajectory planning. Eventually, the data could also be used to calibrate the internal inclinometer or any other auxiliary inclinometer used in a campaign. The SSL is performed regularly to check if the alignment has changed.

To assess the robustness of the SSL and its performance against the internal inclinometer of the scanner system, we ran the SSL continuously for almost 18 days from 6–24 August 2020. Figure 6(a) and (b) show, respectively, the time series of tilt

and roll obtained from the SSL and the internal inclinometer. Each time step represents the result of a full SSL scan (with a duration of about 2.5 min) and the corresponding mean value of the inclinometer data. In addition, the standard deviation of the inclinometer is shown as a band of $\pm\sigma$. The results show a good correlation between the two signals. The SSL indicates a different bias in each axis of the inclinometer, namely $tilt_{bias} = -0.05°$ and $roll_{bias} = +0.05°$. A change in mean tilt and roll over time can be observed. This is due to the varying conditions of the support structure. In particular, the thrust of the wind

turbine changes the magnitude and direction of the tower inclination depending on the wind direction and wind speed. The error between the two sensors (now assuming SSL as the sensor) can be seen in Figure 6(a) and (b). After debiasing both errors, we obtain mean-square errors of $\epsilon_{tilt} = 0.01°$ and $\epsilon_{roll} = 0.02°$. Finally, the variance of the inclinometer is a consequence of the system dynamics, which must be taken into account when assessing the accuracy of the scanner alignment. The total variance of the inclinometer signals is $\sigma_{tilt} = 0.23°$ and $\sigma_{roll} = 0.24°$. It should be noted that in the absence of information on the



calibration of this sensor, we assume this value to be conservative. This is based on the assumption that the sensor not only perfectly detects rotational changes, but that the resulting values are a superposition of rotational and translational movements.

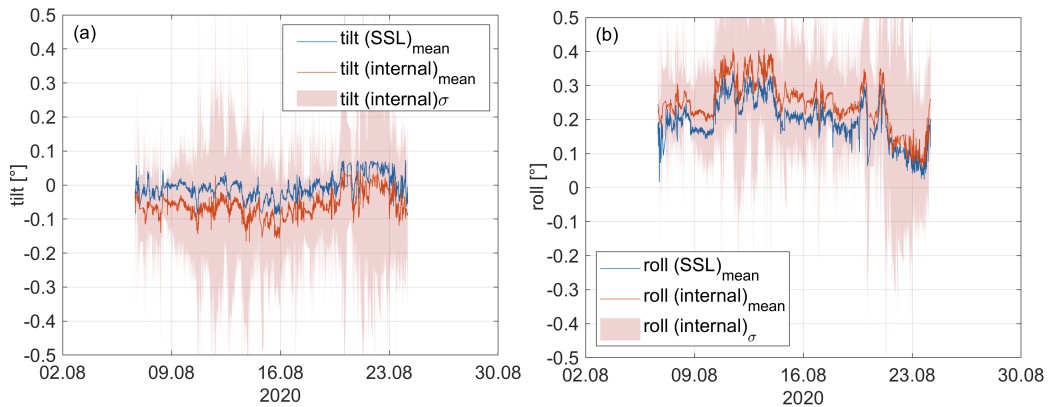

**Figure 6.** (a) Lidar tilt from internal inclinometer (internal, red) and from SSL (blue). Red band shows the standard deviation (std dev) of the inclinometer tilt during each SSL scan. (b) Lidar roll from internal inclinometer (internal, red) and from SSL (blue). The red band shows the standard deviation ($\sigma$) of the inclinometer roll during each SSL scan.

## 2.2 Mesoscale model

Mesoscale simulations, including both undisturbed free wind data and wake-induced wind data due to the surrounding wind farm, were performed using the WRF model (version 4.2.1) developed by the National Center of Atmospheric Research (Ska-
marock et al., 2019). In the WRF model, there are prognostic variables for the horizontal and vertical wind components, potential temperature, geopotential and surface pressure of dry air as well as several scalars such as cloud water and water vapour. The WRF model is well known and widely used in the wind energy community (Hahmann et al., 2020; Kibona, 2020), and in recent years also for wind farm wake simulations (Pryor et al., 2019; Siedersleben et al., 2018b).

Our set-up was optimized within several research projects for wind energy applications, especially with a focus on offshore
conditions (Gottschall et al., 2018; Dörenkämper et al., 2020; Gottschall and Dörenkämper, 2021). The studies by Gottschall et al. (2018) and Gottschall and Dörenkämper (2021) compare the mesoscale model data from a similar set-up against vertical lidar and mast measurements. To limit the number of grid points in the numerical calculations, a nesting technique is used. Three domains centered around the German Bight area are nested, each of a size of 120 grid points with resolutions of 18 km, 6 km and 2 km. Figure 7(a) shows the distribution and size of the three domains around the site of interest.

The wind turbines were parametrized as momentum sinks and source of turbulence using the Fitch wind farm parameterization (Fitch et al., 2012). In every grid that intersects the rotor disk, the horizontal wind component is reduced to represent the drag of the wind turbine. Different thrust and power curves corresponding to all turbine types were applied. Figure 7(b) shows the locations of the turbines in the model simulations. Boundary conditions for the model were prescribed by the ERA5 dataset (ERA5 resolution, $0.25° \times 0.25°$ ($\sim$30 km), 6-hourly) for the atmospheric variables (Hersbach et al., 2020) and the OSTIA



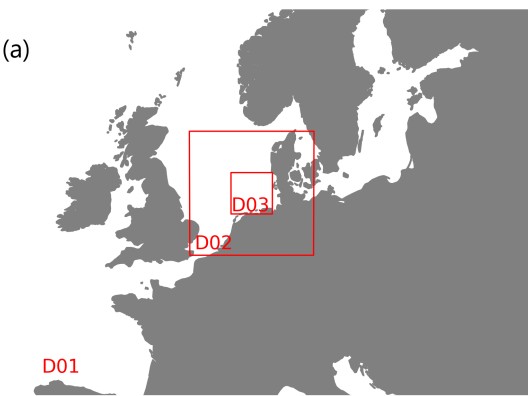 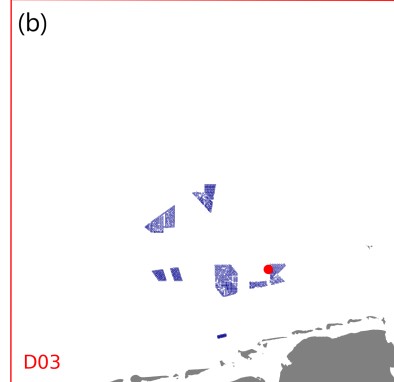

**Figure 7.** WRF model domains (D01, D02, D03) and a magnification of on the innermost model domain (D02: (a) locations of the three model domains with a grid sizes of 18, 6 and 2 km, respectively. The innermost domain of (a) is shown in detail in (b) with the locations of the wind turbines accounted for in the simulations and the location of the lidar marked in red. Note that wind farms with a distance of more than 100 km from the site were ignored.

dataset for the sea surface variables (Donlon et al., 2012), which provides near-real time global sea surface temperature at the grid resolution of $1/20°$ ($\sim$6 km). The WRF version used in this study does account for the turbulent-kinetic-energy advection bug that was recently discovered (Archer et al., 2020).

We performed simulations with and without wind farms and extracted the time series from the WRF simulations at the position of the scanning lidar measurements for the same period. The most important settings of the mesoscale model configuration 220    are summarized in Table 4.

### 2.3 Airborne measurements

For investigating the spatial extent of wakes generated by wind farms, several airborne measurements were carried out within the scanning wind lidar measurement period using the research aircraft Dornier DO-128 operated by the Technische Universität 225    Braunschweig (Lampert et al., 2020). The airborne measurements carried out simultaneously are an ideal complement to the scanning lidar measurements. They enable the analysis of various atmospheric parameters like temperature profiles, as well as wind direction and speed over a larger area. Although the data are limited to a few hours and days, they provide an overview of the wind situation at different distances and heights both upstream and downstream of the wind farm clusters. The research aircraft is equipped with a nose boom to perform high-resolution measurements of the wind vector, temperature 230    humidity and pressure, sampling at a frequency of 100 Hz (Corsmeier et al., 2001). Further, a sensor for measuring the surface temperature, a laser scanner for determining sea state characteristics and cameras were integrated (Lampert et al., 2020). During the measurement flights, both the upstream and downstream areas of the wind farm clusters were investigated. The flight pattern included legs of 45 km length that were aligned perpendicular to the main wind direction, therefore crossing the wakes, and





**Table 4.** Relevant parameters of the mesoscale model set-up. The references for the different schemes and models are summarized in WRF Users Page (2020).

| Parameter | Setting |
|---|---|
| WRF model version | 4.2.1 |
| Planetary Boundary Layer (PBL) scheme | MYNN level 2.5 |
| Wind farm parametrisation | Fitch et al. (2012) |
| Land-use data | MODIS |
| Surface-layer scheme | MYNN |
| Microphysics scheme | WRF Single-Moment 5-class |
| Shortwave and long-wave radiation | RRTMG |
| Atmospheric boundary conditions | ERA5 |
| Sea surface conditions | OSTIA |
| Horizontal resolution | 18 km, 6 km, 2 km |
| Vertical resolution | 60 eta-level |
| Nudging | grid nudging above PBL |
| Model output interval | 10 min |
| Nesting | one-way |
| Land-surface model | Unified Noah Land Surface Model |
| Simulation duration | 240 (+24 spin-up) hours |

vertical profiles from around 15 m altitude up to 1000 m. The individual straight flight legs were horizontally spaced about
10 km from each other. The measurement height was 120 m above sea level, which corresponds to the hub height of the wind farms Gode Wind 1 and 2.

An example of a flight dataset showing multiple wake-transect profiles perpendicular to the mean wind direction, measured downstream of clusters N-2 and N-3 on 3 July 2020, is briefly presented in the next section.

## 3    Wind field modification by wind farm clusters

The strong modification of the wind field by the wind farm clusters is clearly evident in scanning lidar measurements, flight measurements, and WRF simulations. Flight measurements enable an initial side-by-side evaluation of the WRF model over a larger spatial scale than is possible with just the scanner lidar system, and give a first qualitative impression of the strength and extent of wind farm cluster wakes (Sect. 3.1). The lidar measurements are then compared with WRF model results with and without a wind farm parameterization for the different sectors and atmospheric stability conditions, which enables an
evaluation of WRF performance for different upstream wind conditions (Sect. 3.2).



## 3.1 The spatial extension of cluster wakes

Simultaneous flight measurements may be used to complement the lidar measurements because of the extended range of flight paths around wind farm clusters, but only for short periods of time. We consider here the wake situation of the N-2 and N-3 clusters on 3 July 2020 (10:24–13:02 UTC) when flight legs were flown perpendicular to the wind direction ($\approx 230°$) and

taking on average 10 min per traversal (see the orange lines in Figure 8(a)). The spatial distribution of the measured wind speed is inferred in Figure 8(a) by linearly interpolating the flight legs in the wind direction. Darker colors, representing lower wind speeds, are evident directly behind wind farms and more dense clusters of wind turbines. In particular, the strong reduction in wind speed caused by cluster N-2, which is located to the west, but also by the wind farms Gode Wind 1 and 2, which are located further to the east, can be clearly seen. Behind the north-eastern edge of Gode Wind 2, the wind speed is about

$7.5\,\mathrm{m\,s^{-1}}$. Upstream of the wind farm, the wind speed is about $11\,\mathrm{m\,s^{-1}}$, which corresponds to a reduction in wind speed of about 30%. Stability during this period was inferred from vertical temperature profiles outside the wake area influenced by the wind farm clusters by steep climbing and descending flight profiles up to an altitude of about 1000 m, which reveal, after an initial close-to-neutral period ($-0.005\,\mathrm{K\,m^{-1}}$), stable conditions (with a maximum value of $0.01\,\mathrm{K\,m^{-1}}$) for the last legs and thus explain the strong wakes detected.

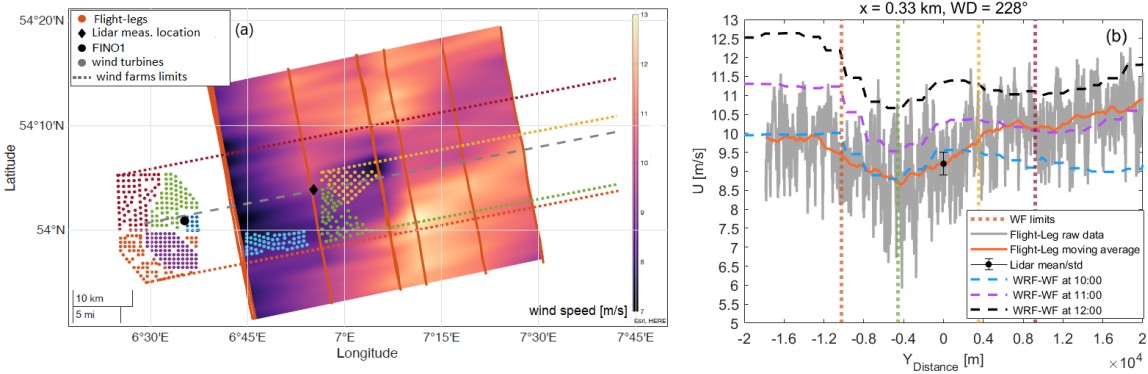

**Figure 8.** (a) Spatial distribution of the measured wind speed from the measurement flight on 3 July 2020 (10:24 – 13:02 UTC). The data are linearly interpolated in the mean wind direction. The colored points represent the wind farm (WF) limits, the orange solid lines show the locations of the flight legs, and the dashed lines indicate the downstream boundaries of the clusters for this wind direction. (b) Horizontal wind speed profile for the closest flight traversal upstream of the lidar measurement point ($x = -0.33\,\mathrm{km}$) and perpendicular to the nominal wind direction of $230°$. The N-2 and N-3 cluster wake limits are indicated by dashed lines. The gray line represents the 100 Hz data, the orange line shows the data filtered by a moving average, and the black diamond with error bars represent the lidar mean wind speed and its standard deviation for the duration of the flight-leg. The dashed blue, purple, and black lines show the WRF results with a wind farm parameterization (WRF-WF) from a transect of the model results at 10:00, 11:00, 12:00 UTC, respectively, based on the flight coordinates.

Figure 8(b) shows the horizontal wind speed profile for the closest flight traversal (orange line) upstream of the lidar measurement point ($x$ = -0.33 km) compared with the mean wind speed according to the lidar (black diamond) and standard deviation





for the duration of the flight leg (error bars), revealing a suitable agreement between the two measurements for this flight (mean bias = 0.06 m s$^{-1}$). For this leg, the lapse rate was -0.003 K m$^{-1}$, which explains the relatively high turbulence signal in the 100 Hz data (gray line). The flight-leg was performed between 11:27 and 11:37 UTC. For comparison, the WRF-WF model results for the times 10:00, 11:00, and 12:00 UTC are shown for a transect corresponding to the flight coordinates. Note that for this particular time period, there is an approximate phase error of approximately 1 h, which is common in WRF results.

The flight altitude corresponds to approximately 120 m, corresponding to the mean hub height of the N-2/N-3 clusters and the mean wind direction of 228°. The blue curve in Figure 8 represents data extracted from the WRF model with the wind farm parameterization at the position of the flight leg. While these data also show a decrease in the wind speed between the limits of the cluster N-2, the wake minimum is about 0.5 m s$^{-1}$ higher than the flight-leg. The mean wind direction difference between the flight-leg and the simulation is about 10°. It is worth mentioning that the WRF simulation does not take into account the wind farms located at about 15 km east of the cluster N-2 (Gemini wind farm), which could explain the difference of more than 1 m s$^{-1}$ at the northern part of the wind farm wake (negative $x$-axes).

The freestream wind speed is estimated assuming a linear horizontal background wind gradient along each traversal based on the wind speed outside the wake limits (outer dashed lines). The 100 Hz data are filtered by a moving average with a window size of 10-min; its purpose is to filter turbulent fluctuations that would otherwise bias the shape of the fitted wind speed to regions of higher turbulence (orange line).

Figure 9 shows the WRF model simulated horizontal wind field with the wind farm parameterization at hub height (120 m) and at different time steps, illustrating that the spatial dimensions of the modeled and observed wakes agree well. Both the observations (Figure 8) and model (Figure 9) show a wake extending at least 40 km downstream of the N-3 wind farm cluster, meaning this wake was long enough to reach the wind farm cluster N-4 (not shown) located about 60 km downstream of Gode Wind. In the spanwise direction, the wake has dimensions of approximately the maximum width of the wind farms. The simulations for different time steps indicate well the temporal variability of the wind field, which has to be considered for a flight of 4 h duration as well. As shown next, significant wake effects were detected by the scanning lidar for cases such as these for flow from the east.

## 3.2 Directional and stability dependence of cluster wakes

We evaluate the lidar-derived wind speed measurements by first dividing the wind direction into five unequal sectors within the cluster wakes (see Figure 1). The lidar measurements are then compared with mesoscale model results with and without a wind farm parametrization for the different sectors, which enables an evaluation of the model performance for the different upstream conditions. Note that because the stability is also wind direction dependent at this site, some sectors are affected more by certain stability conditions. Figure 10(a) shows the wind rose from the lidar measurements together with the five wind direction regions R1–5, illustrating the predominance of a west-south-westerly wind direction for the data period presented here, which corresponds to the main meteorological wind direction in the German Bight area (Cañadillas et al., 2020). Figure 10(b) shows the Weibull distribution for the wind speed together with the scale parameter $A$ and shape parameter $k$ for each region R1–5 at 120 m. This illustrates that meteorological conditions and wind speed distributions within a region are very



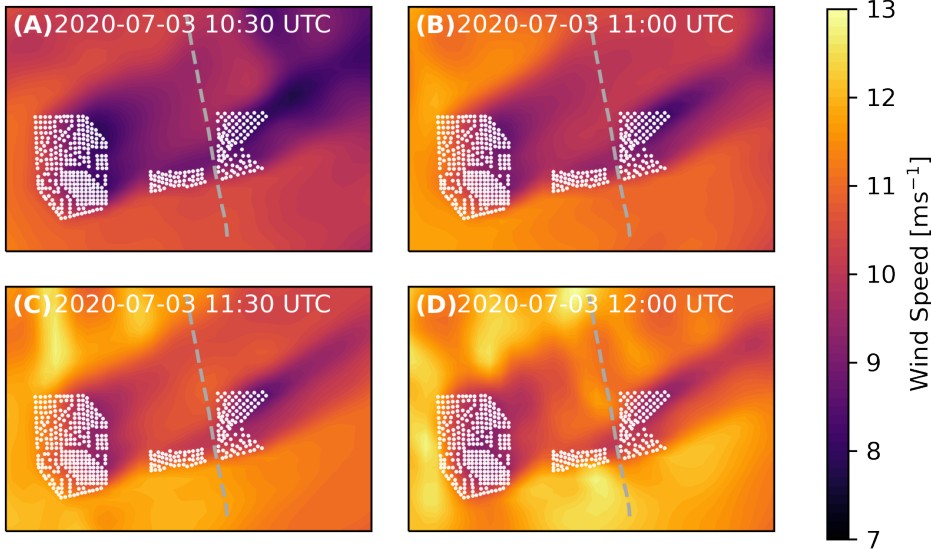

**Figure 9.** Spatial distribution of the measured wind speed (see colorbar) from the WRF simulation on 3 July 2020 at different timesteps. The dashed line indicates the position of the flight-leg shown in Figure 8

different, so that a direct comparison of wind data between the different sectors does not make sense due to the different flow conditions found in each sector.

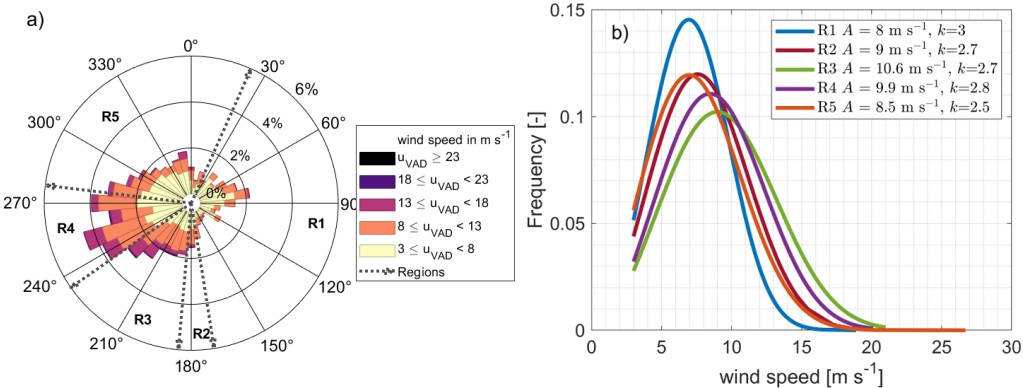

**Figure 10.** Wind rose (a) and Weibull distribution per regions R1–5 (b) at 120 m altitude measured by the scanning wind lidar for the period May to September 2020.




The wind roses derived from lidar data for different atmospheric stabilities are shown in Figure 11, illustrating that a large part of the data obtained during stable atmospheric stratification corresponds to flow from the east. With neutral atmospheric

stability, southwesterly winds prevail, while in unstable stratification northwesterly winds are predominant. Unstable and neutral conditions are associated with higher average wind speeds than for stable conditions.

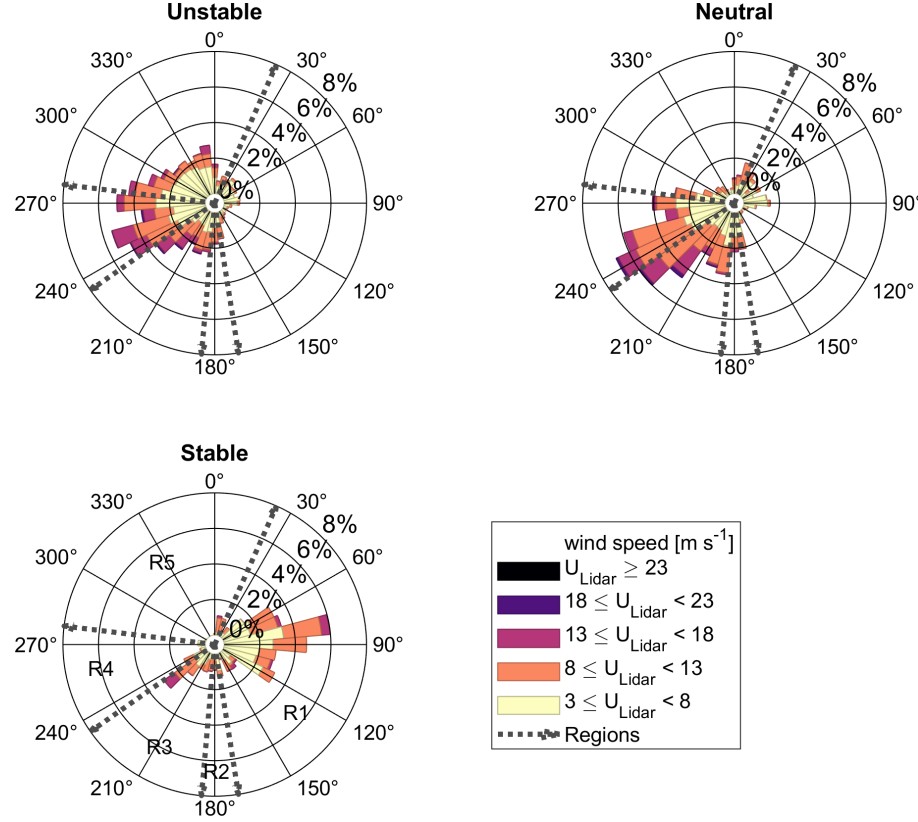

**Figure 11.** Wind roses measured by the scanning lidar for different atmospheric stabilities at 120 m above LAT. The regions R1–5 are indicated by dashed lines.

The wake-induced wind speed reduction at the position of the lidar measurements is investigated for each wind region using a polar plot for which the mesoscale simulation results (without wind farm parameterization) are used as the reference free wind speed. Since the model data represent an undisturbed state not influenced by wind farms, differences in the wind

conditions are to be expected when comparing the two datasets. Especially in the regions R1, R3 and R4, which are directly influenced by the wind farm clusters, lower measured wind speeds are expected. Figure 12 shows the direct comparison of the wind speeds per wind direction of the lidar and the mesoscale model data at 120 m height for (a) unstable, (b) neutral, and (c) stable atmospheric conditions. For better understanding, the boundaries of the regions R1 to R5 and the individual positions of the wind turbines are also indicated.


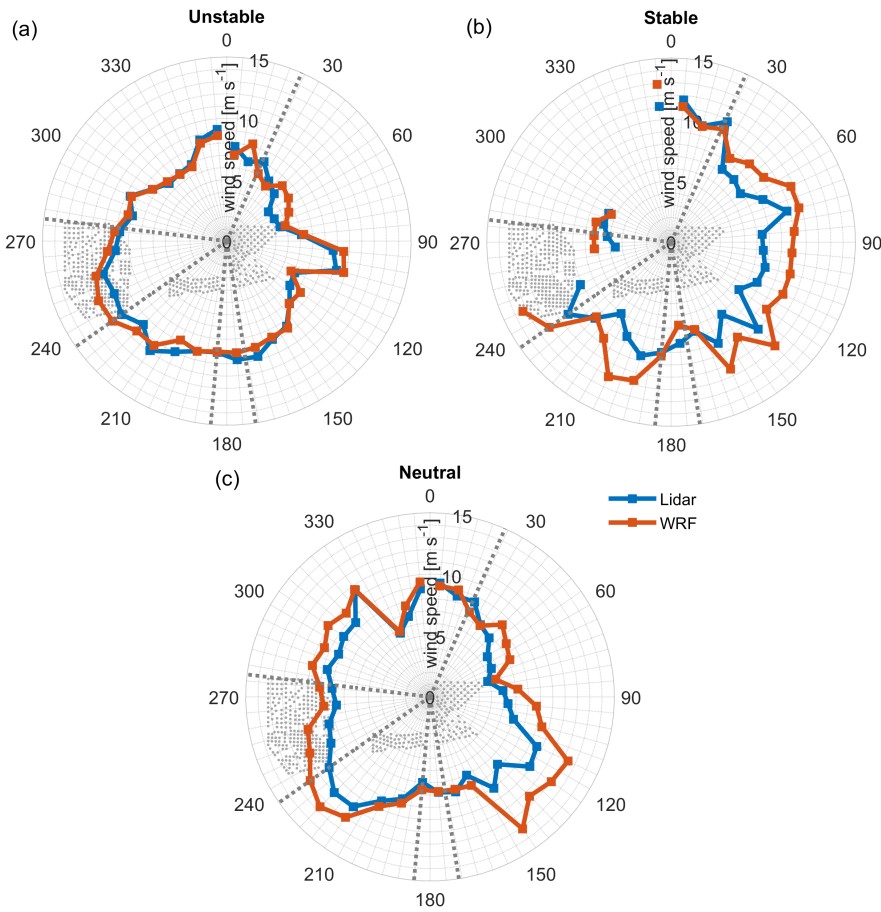

**Figure 12.** Wind speed polar plots of the lidar measurements and WRF results (without the wind farm parameterization) for (a) unstable, (b) stable, and (c) and neutral atmospheric stratification at the height of 120 m. Wind turbines are indicated by gray points, and the regions R1–5 by dashed gray lines.

Under both unstable and neutral atmospheric stratification, the wind speed distributions of both datasets show good agreement, especially in regions R5 and R2, the free-wind areas. The difference in wind speed in the other regions is also minimal under unstable stratification where the influence of the wind farms is difficult to detect in the measurements since wind farm wakes are not expected to be large. In contrast, a comparison of the datasets for neutral and stable atmospheric stratification shows a clear discrepancy between the measurements and model results, with this difference particularly evident in the regions R1, R3 and R4, which are directly influenced by wind farm clusters. The maximum difference is about 4 m s$^{-1}$ in region R3 in stable stratification. Even with neutral stratification, a difference can be seen in the two wind speed datasets and the regions influenced by wind farms. In the free-wind region, however, both datasets agree very well. The strong fluctuations in the wind speed of the lidar data in region R1 are due to the very small amount of data in neutral stratification (see Figure 12). For region R5, in the case of stable atmospheric stratification, there is no or too little data available in some wind direction sectors. A





comparison of the data from the free-wind sector is not possible here. Nevertheless, the reduction in wind speed caused by the wind farms in the other regions can be clearly seen.

To quantify the effects of the wind farm wake in the different regions, the wind speed difference $u_{diff}$ of the lidar-measured
wind speed $u_{lidar}$ with respect to the mesoscale wind speed $u_{WRF}$ is defined as

$$u_{diff} = \left( \frac{u_{lidar}}{u_{WRF}} - 1 \right) 100\%, \qquad (2)$$

and presented in Figure 13 as histograms of the averaged wind speed difference for each region in unstable, neutral, and stable conditions. The bars within a group represent the five measurement heights of 40 m, 80 m, 120 m, 160 m and 200 m, and the number of 10-min lidar values within a bar is shown at the end of the bar. The results from the WRF model without the wind
farm parameterization $u_{diff}$(WRF) and the stationary lidar measurements in region R5, the free-wind region, show a relative good agreement for neutral and unstable conditions ($u_{diff}$(WRF) = 5% and 10%, respectively), and especially for unstable stratification (1–2%), for all measured heights. For the second free-wind range R2, only small differences between the WRF model and lidar data are evident in the case of unstable and neutral atmospheric stratification. However, the results in this range vary strongly with height. One possible reason for this is that the lateral extent of the narrow undisturbed corridor in region
R2 is too small, only about 3 km, and that the boundaries of the wake effects become wider with increasing distance to a wind farm due to the wake expansion. This effect is enhanced by a stable atmosphere. Even when the corridor was further narrowed by changing the region boundaries of R2, no effect similar to R5 was detected. The small horizontal extent of the corridor and the large distance of the lidar measurement site from this area make a differentiated evaluation of region R2 difficult. The regions R1, R3 and R4 influenced by the wind turbines all show a relatively large wind speed deviation of the lidar data from
the WRF results without the wind farm parameterization. As expected, the values measured are lower than the values from the undisturbed computational model at all heights and stability conditions, with the attenuation especially pronounced in stable stratification. In region R4 and at a measurement height of 120 m, a reduction of the wind speed by about 26% can be seen in stable stratification. It is also noticeable that the reduction in wind speed shows a negative trend with increasing height. While the maximum height of the lidar measurements may be 200 m and most of the wind turbines in the surrounding area have a
total height of between 140 m and 180 m, some interaction effects can also be detected above the wind farm due to vertical wake expansion (Siedersleben et al., 2018a; Larsén and Fischereit, 2021). However, as measurements at a height of 200 m are only partially influenced by the wind turbines, as they are no longer completely behind the rotor surface, this probably explains the lower wind speed difference $u_{diff}$(WRF) at 200 m. The wind speed differences can be further emphasized for different atmospheric stability conditions within a region. Here it is noticeable that, in regions R4 and R3, a strong reduction in wind
speed behind the wind farms with increasing atmospheric stability can be seen but this is much less pronounced in region R1. As regions R3 and R4 have a larger distance to the measuring point of the lidar than region R1, it can be assumed that, in an unstable atmosphere, the wind speed recovers more quickly in the wake of a wind farm than in stable conditions, as also shown in Cañadillas et al. (2020).

The lateral extent of a wind farm, i.e. the number of wind turbines in the flow direction as well as the wind turbine layout, also







**Figure 13.** The difference $u_{diff}$ (Eq. 2) in the wind speed between the lidar measurements and WRF model (top: no wind farm parameterization $u_{diff}$(WRF); bottom: with the wind farm parameterization $u_{diff}$(WRF-WF)) for each region R1 to R5, for each measurement height 40 m, 80 m, 120 m, 160 m and 200 m, and for each stability class. The number below/above each bar indicates the number of 10-min wind speed values.

affects the strength of the change in wind conditions in the wake of a wind farm cluster. Region R3 is influenced by the wind farm Nordsee One. Here, a maximum of six to seven wind turbines are located behind each other in the flow direction. In region R4, which is influenced by the wind farm cluster N-2, the number of wind turbines in the direction of flow is 17 to 20 turbines,



depending on the wind direction. This effect is evident when considering the wind speed differences in regions R3 and R4 under unstable atmospheric conditions. Although the distance of the wind farm in region R3 to the measurement location of
the lidar is significantly smaller, the reduction in wind speed is smaller than in region R4. A possible reason for this is the size of the wind farm cluster in R4. Figure 14 shows a polar plot comparing the WRF outputs, both with and without wind farms, with

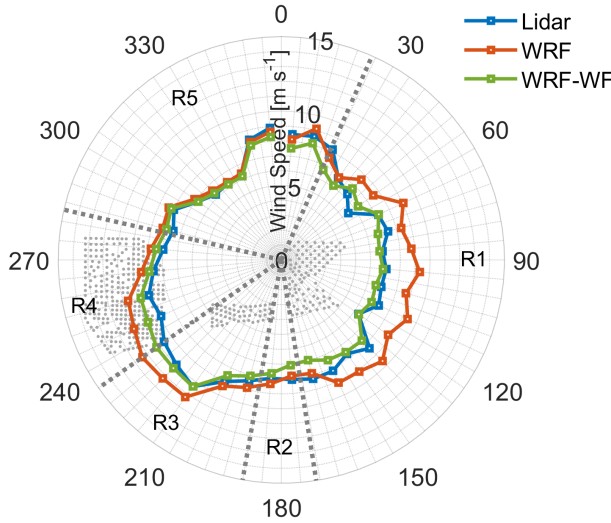

**Figure 14.** Wind speed polar plot of the lidar (blue), WRF model without (red) and with wind farm influences (WRF-WF, green) at 120 m. Wind turbines are indicated as gray points and the sectors are indicated as dashed gray lines.

the lidar, where there is good agreement between the lidar data and the WRF simulation with wind farm influences for both the unstable and neutral atmospheric layers, but not for the stable layers. To ensure a fair comparison between the mesoscale (WRF-WF) model and the lidar, hourly production data from energy-charts (available at: https://www.energy-charts.de/, last
access: July 2021) were used for filtering purposes. Only wind farms in operation at the measurement times are included in the mesoscale simulations and thus considered in the comparison. In general, the model performs reasonably well and is able to capture the wake variation with wind direction. It is especially evident that the mesoscale model simulations with the wind farm parameterizations better reflect the measured wind speeds compared with the standard WRF model.

     The data presented in Figure 14 are divided into unstable (a), neutral (b), and stable (c) conditions in Figure 15. A good
agreement is found for most of the regions (R2, R3 and R5) under unstable conditions with a wind speed differences of around 2% in wind speed. As expected, the larger disagreements in wind speed (almost 20%) are found under stable conditions for the regions R1 and R4 (downstream of the large wind farm clusters Gode Wind/N-2), and of around 10% for the region R3 (downstream of the relatively small wind farm Nordsee One).

     Figure 13 (lower panels) also presents the wind speed difference for the mesoscale model $u_{diff}$(WRF-WF). In general, the
wind farm parameterization reduces the absolute magnitude of the wind speed difference $u_{diff}$ in the waked regions, especially for regions R1 and R4 to, respectively, the east and west of the lidar for all stability classes, and for region R3 to the south-west



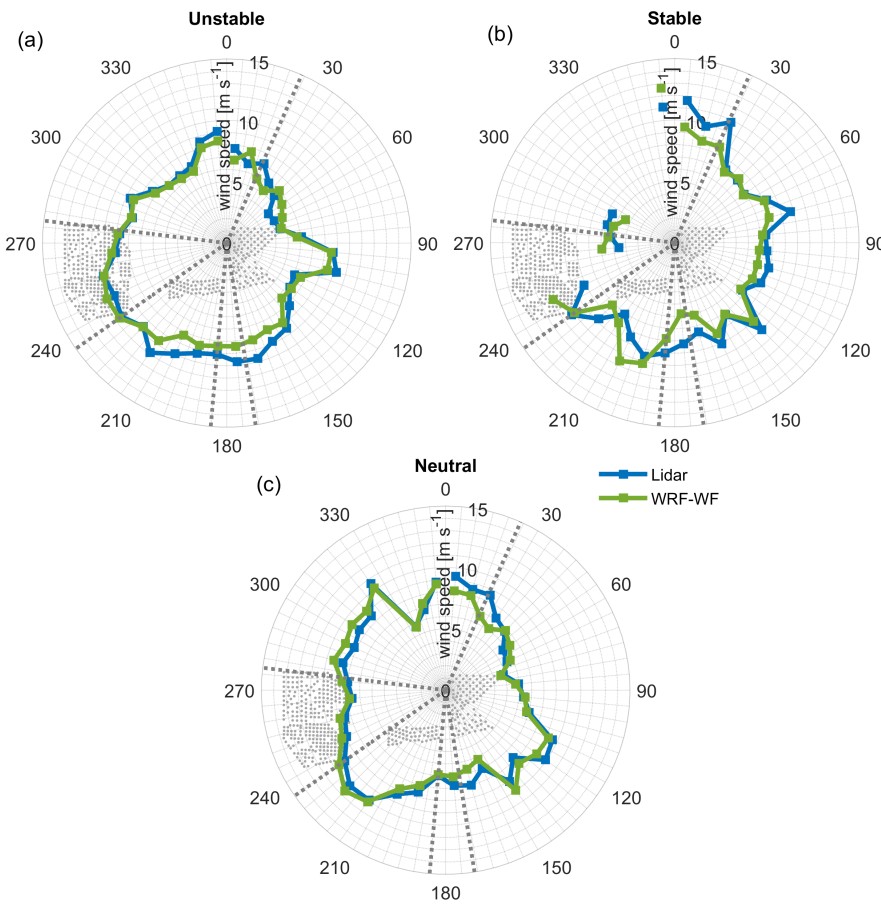

**Figure 15.** Wind speed polar plots of the lidar measurements and WRF-WF results (with the wind farm parameterization) for (a) unstable, (b) stable, and (c) neutral stratification at the height of 120 m. Wind turbines are indicated by gray points, and the regions R1–5 by dashed gray lines.

of the lidar, except for unstable conditions where the value of $u_{diff}$ is more positive for all heights. This could be due to coastal effects to the south not being properly captured by the model (see also the southern part of the polar plot in Figure 15(a)). The difference in the narrow region R2 to the south is also worsened by the wind farm parameterization, including for all stability classes. Our lidar measurements thus serve as a reference for further improvements in wind farm parameterizations.

## 4 Conclusions

Interaction effects between wind farm clusters N-2 and N-3 in the German Bight are demonstrated via the analysis of data from a scanning lidar, airborne campaign, and mesoscale model. Lidar measurements combined with meteorological sensors reveal the strong directional and stability dependence of the wake strength in the direct vicinity of wind farm clusters. For

sectors without upstream wind farms, the scanning lidar data agree with the mesoscale simulations of the undisturbed flow in unstable, neutral and stable atmospheric conditions. In region R5 (sector free of wind farms to the north), the maximum wind speed difference is about 2%, whereas in region R4 a reduction of up to 45% was observed. The magnitude of the difference increased in all other regions with increasing atmospheric stability. The smallest, but still most significant differences were seen at 200 m altitude, above the top of the rotor blades. The wakes still have an influence here, but it is much smaller than

at hub height. This effect is most apparent when examining the wind speed at other measurement heights. This dataset allows for numerical model validations. In this way, numerical simulations can be validated. Taking into account the mesoscale wind farm parameterization (WRF-WF), overall, the model performs reasonably well and is able to capture the wake trend. A good agreement is found for most of the regions (R2, R3 and R5) in unstable conditions with a relative deviation of around 2% in wind speed. As expected, the larger disagreements in wind speed (almost 20%) are found for stable conditions for the regions

R1 and R4, amounting to around 10% for region R3. This means that mesoscale wake simulations still have deficiencies in correctly reproducing the atmospheric stratification and its influence on the development and decay of wind farm wakes. Scanning wind lidar measurements are therefore a powerful tool for the evaluation and improvement of wind model simulations and in particular wind farm parameterizations. Ongoing work within the X-Wakes project aims to combine this analysis with operational data from surrounding wind farms to validate the performance of industry models by focusing on the interactions

of cluster wakes.

*Data availability.* The airborne data will be published in PANGAEA after the end of the project X-Wakes.

*Author contributions.* B.C. wrote the manuscript with the support of R.F. and A.L.. M.B. and B.C. evaluated and prepared most of the figures. J.T. together with B.C. processed the scanning lidar data and wrote the scanning lidar section. M.D. provided the WRF data and the description of the simulation. T.N. was involved in the funding acquisition and supervised the research. All authors contributed intensively

to an internal review.

*Competing interests.* The authors declare that they have no conflict of interest.

*Acknowledgements.* The authors would like to thank Rolf Hankers, Thomas Feuerle, Helmut Schulz and Mark Bitter for coordinating and conducting the flight campaigns. Special thanks go to Alexander Tschirch, Hauke Decker and Richard Fruehmann from UL International GmbH, for the logistics and installation of the scanner wind lidar. We thank the operator of the wind farm Gode Wind (Oersted) for their

support and help with the installation. The X-Wakes project is funded by the German Ministry of Economic Affairs and Energy (BMWi) under grant number FKZ 03EE3008 (A-G) on the basis of a decision by the German Bundestag.



## Appendix A: Quantification of scanning wind lidar uncertainties

Uncertainties are something inherent in any field measurement and are usually specified in known procedures or standards. In the case of scanning wind lidar, no standard has yet been developed, so in this section we attempt to explain the uncertainties that are considered relevant in the context of this study.

### A0.1 Calibration

Before the offshore deployment of the lidar a calibration of the system was performed against a reference measurement. For this campaign the system calibration has been performed following the calibration procedure of conventional vertical profilers (IEC-61400-12-1, 2017) at UL's test field in Wehlens in the North of Germany. The uncertainties obtained in this way are assumed to be conservative if compared to a LOS calibration (Borraccino et al., 2016). This increased uncertainty is due to the large height span and associated wind sheer within a single range gate that results from the steep elevation angle of 60°. During the campaign, the maximum elevation angle was closer to 7° and hence the height range covered by a single range gate is significantly smaller. Accordingly, the uncertainty is expected to be smaller than that obtained during the system verification.

### A0.2 Beam positioning

Another uncertainty component is the lidar laser-beam positioning which describes the combined effect of the accuracy of the scanning head in the vertical direction and the effect of the vertical wind shear. Here we used scanner precision as given by the manufacturer and used a vertical power law profile with an exponent $\alpha = 0.14$.

The mounting error has been quantified by means of SSL and has been taken into account in the scanning trajectory design. In this way we almost diminish this error, however there is still a remaining error. This component is made up of the mean error between SSL and the internal inclinometer, as explained above. The error of the measurement height due to the curvature of the Earth is considered negligible at the range distance of the measurement location (1.5 km).

Height variations in the measurement are caused by orientation (tilt and roll) changes. An analysis of these signals from the internal inclinometer has been performed over the whole campaign. Finally, the effect of wind shear has been evaluated using the vertical power law profile with an exponent $\alpha = 0.14$.

Additionally, as pointed out in Newsom et al. (2017) with respect to the VAD method, deviations from the perfect sinusoidal occur due to spatial and temporal fluctuations in the velocity field and instrumental errors and in the context of the VAD algorithm, any departure from the perfect sinusoidal may be regarded as error. Due the lack of a proper physical set-up during this study, numerical simulations have been performed to assess the robustness of the calculation chain of the partial VAD. The results show an average of approximately 3% mean error due to wind field inhomogeneity on our partial VAD procedure. This can be seen as a conservative estimation of uncertainty in the wind field reconstruction.





The simulations were based on synthetic fields with a defined mean wind speed ($U$) with superimposed Gaussian random noise $\sigma/U = 10\%$. The wind field was scanned with the same geometry as our partial VAD and for all combinations of parameters shown in Table A1.

**Table A1.** Parameters set-up for the VAD simulation.

| Parameter | Range | Step |
|---|---|---|
| Wind speed ($U$) [m s$^{-1}$] | [5, 30] | 5 |
| Wind direction [°] | [0, 360] | 30 |
| Scanning elevation [°] | [1, 9] | 2 |
| Gaussian noise ($\sigma/U$) [%] | 10 | – |

The results in Figure A1 show the root-mean-square error (RMSE) of simulations against the reference mean wind speed for 100 runs of each parameter combination. A dependence of the wind field reconstruction on both the azimuth opening angle of the scan trajectory and the wind speed can be observed. An average value was obtained based on the dependencies and wind speed distribution.

The effect of inhomogeneous flow, mainly caused by partial wake effects, has been evaluated based on simulation results from Lundquist et al. (2015).The value is based on the assumption of a wake distance of nine rotor diameters downstream.

**Table A1.** Summary of uncertainty components that contribute to the global uncertainty of the wind measurement with the scanner lidar and analysis techniques used during this project

| Component | Estimated value [%] | Remark |
|---|---|---|
| Calibration | 1.4-2.2 | Conventional calibration as a vertical profiler |
| Beam positioning | 0.1 | Based on scan head accuracy |
| Mounting error | 0.1 | Estimated from SSL |
| Orientation dynamics | 1.7 | Conservative value from inclinometer variance over the measurement period |
| Wind speed reconstruction | 3.0 | Conservative value obtained from simulation of our partial VAD. |
| Wind field inhomogeneity | 2.5 | Conservative value applicable for situations in wake. Obtained from Clifton et al. (2018). |



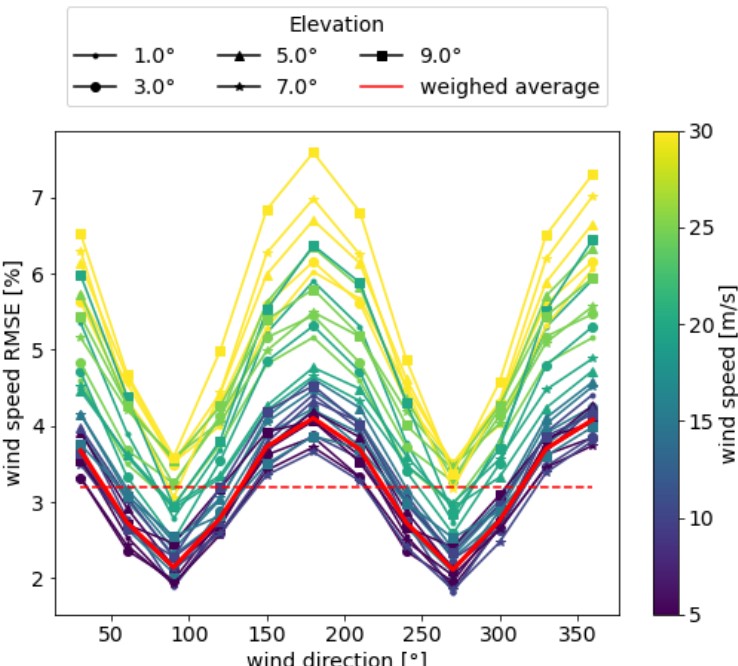

**Figure A1.** RMSE of simulated VAD. The resulting RMSE is calculated for 100 runs of each combination of parameters as shown in Table A1. Lines in red represent values weighed with the frequency distribution of wind speed and wind direction at the site. The dashed line represents the average of the continuous red line.

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
