# Peer review of "Offshore wind farm cluster wakes as observed by long range scanning wind lidar measurements and mesoscale modeling"

_Wind Energy Science, 2021_

## Referee Comment (RC1)

**REVIEW OF WES-2021-159**

*Offshore wind farm cluster wakes as observed by a long-range scanning wind lidar*

*authors:*
Beatriz Cañadillas,
Maximilian Beckenbauer,
Juan J. Trujillo,
Martin Dörenkämper,
Richard Foreman,
Thomas Neumann,
Astrid Lampert

**Summary:**

The manuscript entitled "Offshore wind farm cluster wakes as observed by a long-range scanning wind lidar" presents data collected with a scanning lidar during an offshore field campaign and compares to WRF simulations with and without the wind farm parameterization. The comparison shows very good agreement between model results and observations for most wind direction sectors and atmospheric stability conditions, which helps to establish the reliability of WRF predictions. However good the comparisons, the ultimate goal of the paper remains somewhat unclear. If the authors intend to perform validation of the WRF-WFP, there needs to be much more detail regarding the implementation of the model, known points or sources of uncertainty, and validation methodology. If the authors are creating a public data set for other researchers to use in future validation studies, much more detail needs to be included about data collection, quality control, and access to raw instrument data. Offering clear messaging about the purpose of the publication will help. Otherwise, the paper parses as, "WRF with the WFP matches lidar measurements better," which is not a novel contribution. A discussion section might also help the authors clarify what results are important, and what the analysis contributes to the wind energy, mesoscale modeling, and remote sensing communities.

**Comments:**

- Abstract — Probably "quantify" or "characterize" wakes and wake impacts, in addition to simply detecting them. This sentence would be a good opportunity to tell readers what to expect in your work. The abstract also indicates that airborne measurements are compared to lidars, which doesn't seem to occur in the text.
- Abstract— Error is noted as 30% in the abstract, but 45% in the conclusions. Is this inconsistent or are the authors discussing different points?
- page 1 — "which can be transferred to other regions." The meaning of this phrase is not clear.
- page 2 — "the first phase of a potential offshore wind farm to assess accurately the wind resource," preconstruction or resource assessment phase?

- page 6, line 127 — It is not clear what these coordinates imply. Azimuth and elevation? The unobstructed azimuth angles visible to the lidar (i.e., everything not within the sector $20°$ to $160°$)? If this is the case, how are lidar measurements shown in the figures below obtained for this sector?
- Figure 8 — Linear interpolation across such a large range is sure to introduce uncertainty to the results, particularly near the wind plans. How do these measurements compare to the WRF model results? Are the green and yellow dashed lines relevant to the information in Figure 8b? Since those wind farms are downstream of the flights, I'm not sure what they communicate. Also, the spacing of dashed lines indicating wind farm clusters is very different from Figure 8a to 8b. Any idea why this is if they are showing measurements along the same transect? "...the closest flight traversal upstream of the lidar measurement point $x = -0.33$ km)" Can this be made more clear in the figure? I believe that this refers to the line closest tp the black diamond in Figure 8a. Is that correct?
- page 14, end of Section 3.1 — How are the different scales reconciled. WRF, Lidar, and the instrumentation on the aircraft all make measurements at significantly different scales and resolutions—effectively observing different phenomena.
- Figure 9 — Showing scanning lidar returns would help to understand how the measurements and model results compare. Please show selected scans, or time averages, and give some indication of how much data are collected, measurement uncertainty, etc.
- Figure 10 — Please show histograms that data are fit to and fit quality. Also, R2 contains fewer observations than the other sectors. How does this impact the fit quality or the analysis?
- Figure 11 — Unstable conditions look to be bimodal, from WNW and WSW. Both peaks are captured in R4, indicating that there may be problems with statistical convergence in teh other sectors. Observations during neutral conditions are spread over 2 sectors. Does this impact the quality of the analysis?
- page 18 — "Nevertheless, the reduction in wind speed caused by the wind farms in the other regions can be clearly seen." The polar plots in Figure 12 are very interesting (I love the maps of the wind farms in the background) but are difficult to interpret. Were the WRF simulations really long enough to provide an estimate of wind speed for all the stability conditions and wind direction sectors? Do the lines represent mean wind speeds? Are these the wind speeds at the lidar location or where the lidar scans at 120 m? At what elevation angle? The influence of wind plant wakes is also less pronounced than indicated in the text. Other than perhaps in the stable case (and only for R3), wind plant wakes are difficult to see in Figure 12.
- Equation (2) — It is unclear whether this metric quantifies error in a time series of velocity (i.e., how closely does WRF match the observed wind speed as a function of time), error between average velocities, stability-binned observations, etc. Please describe the error metric in more detail. Also, this definition casts the WRF simulation as the reference signal. How is uncertainty in the WRF simulations included in the analysis?
- page 18 — The two error metrics might be more easily understood as $\varepsilon_{\mathrm{WRF}}$ and $\varepsilon_{\mathrm{WRF\text{-}WF}}$.
- Figure 13 — It is difficult to understand what is important in this figure. Why does WRF-WF under or over predict the wind speed in certain sectors and heights? How is the reader to understand where the measurements are made (e.g., at the lidar location, at XX distance along the line of sight, etc.)? In the caption, please note as (a) and (b). Each subfigure has a top row.
- page 21 — Please include a discussion section. It is difficult to tell from the text which results are important, and what the authors are trying to demonstrate. What is the importance of the study?
- page 22 — "The smallest, but still most significant differences..." What makes them most significant? Statistically significant? Is significance tested?
- page 23, line 431 — This is unclear, do the authors mean, "... in this way we can reduce, but not eliminate, measurement error."?

---

## Referee Comment (RC2)

**Review of the manuscript wes-2021-159, entitled "Offshore wind farm cluster wakes as observed by a long-range scanning wind lidar", by B. Canadillas, M. Beckenbauer, J.J. Trujillo, M. Dorenkamper, R. Foreman, T. Neumann, A. Lampert.**

This manuscript focuses on the analysis of the wind velocity field evolving among various offshore wind farms over the German Bight. This study is performed using data collected with a scanning lidar, airborne measurements, and WRF simulations.

After an introduction, which seems not to have a sharp focus and rather reviewing general offshore wind farm work, the site of the field experiment is described. The section I prefer the most of this work is Sect. 2.1, where the collection of the lidar data is described with a very good level of detail. Subsequently, the WRF simulations are described, in my opinion not providing all the details needed to reproduce this case study, followed by a brief non-technical description of the airborne measurements. The analysis of the data is reported in Sect. 3, starting from a general comparison between the three data sets (Fig. 8), then followed by a more detailed comparison between lidar and WRF data (Figs. 12-15).

I commend the great work done by the authors, especially for the collection of lidar and airborne data. In my opinion, the title of this manuscript is misleading, indeed the reader is going to expect a detailed analysis of state-of-the-art lidar scans to investigate the wind field between wind turbine arrays, while the main focus of the work is an intercomparison between lidar and WRF wind speed at hub height for different wind directions and atmospheric stability regimes, which of course has a more limited scientific interest. Therefore, I am a bit confused about the main insight that this manuscript would convey. At this stage, this manuscript reads more as a technical report rather than a scientific paper. The novelty of this work, if any, should be better emphasized in the manuscript, and discussed in detail. Some other comments are reported below:

1.      L 110-115 - How the various atmospheric stability classes are defined based on the lapse rate? Please provide references as well.
2.      Fig. 8a - I am not sure it makes sense to generate a color map from the linear interpolation of the data collected over the transects. I think it would more informative to show the map with the transect locations and overlap the wind data with linear plots.
3.      Fig. 10 b - I would plot the experimental probability density functions of the wind data, then overlap the respective Weibull distribution and/or the Weibull factors, as reported in the legend.
4.      L 318 – "*The strong fluctuations in the wind speed of the lidar data in region R1 are due to the very small amount of data in neutral stratification*" Can you try to quantify the accuracy of the data through any statistical approach, e.g. error on the mean or percentiles obtained through bootstrapping?

---

## Author Comment (AC1)

**Manuscript WES-2021-159**

**File: wes-2021-159-RC1-supplement%20.pdf**

*We would like to thank reviewer RC1 for the careful reading of the manuscript and for the fair and constructive remarks. We have incorporated your advice into the revised manuscript resubmitted to WES. Below, the reviewer's comments are in bold black and our direct responses to the comments in italics.*

*Beside the current file for the answers to the reviewer, an additional pdf file is attached which refers to the new pdf version of the paper after considering the reviewer's comments. Our changes to the manuscript are additionally presented in blue.*

**General comment**

**The manuscript entitled "Offshore wind farm cluster wakes as observed by a long-range scanning wind lidar" presents data collected with a scanning lidar during an offshore field campaign and compares to WRF simulations with and without the wind farm parameterization. The comparison shows very good agreement between model results and observations for most wind direction sectors and atmospheric stability conditions, which helps to establish the reliability of WRF predictions. However good the comparisons, the ultimate goal of the paper remains somewhat unclear. If the authors intend to perform validation of the WRF-WFP, there needs to be much more detail regarding the implementation of the model, known points or sources of uncertainty, and validation methodology. If the authors are creating a public data set for other researchers to use in future validation studies, much more detail needs to be included about data collection, quality control, and access to raw instrument data. Offering clear messaging about the purpose of the publication will help. Otherwise, the paper parses as, "WRF with the WFP matches lidar measurements better," which is not a novel contribution. A discussion section might also help the authors clarify what results are important, and what the analysis contributes to the wind energy, mesoscale modeling, and remote sensing communities.**

*We agree with the reviewer that the focus of the paper may not be clearly stated in the abstract. Our main focus is on the in situ measurement of an offshore wake cluster with a scanner with lidar, which is a rather new application for this lidar device and not so much literature exists so far. Our intention in this paper is not to perform a thorough validation of the WRF model results, but to use the WRF model primarily as a reference for the free wind, since no measurement of free wind was available during this campaign. It is indeed the case that in the last part of the paper we compare WRF wake clusters with the in situ measurement, but it should be considered as an example of a possible use of such a valuable scanner lidar data. On the other hand, it also helps us to check to some extent the wind reconstruction applied to the original data (namely, the so-called line of sight wind speed).*

*Since this scanner lidar campaign was conducted as part of an ongoing research project (X-Wakes), it is expected that the data will be available by direct request to the corresponding author.*

**Abstract — Probably "quantify" or "characterize" wakes and wake impacts, in addition to simply detecting them. This sentence would be a good opportunity to tell readers what to expect in your work. The abstract also indicates that airborne measurements are compared to lidars, which doesn't seem to occur in the text.**

*Now we have updated the summary, as recommended by the reviewer, as follows:*

*As part of the ongoing X-Wakes research project, a five-month wake-measurement campaign was conducted using a scanning lidar installed amongst a cluster of offshore wind farms in the German Bight. The main objectives of this study are (1) to demonstrate the performance of such a system and thus quantify cluster wake effects reliably and (2) to obtain experimental data to validate the cluster wake effect simulated by the flow models involved in the project. Due to the lack of free wind flow for the wake flow directions, wind speeds obtained from a mesoscale model (without any wind farm parameterization) for the same time period were used as a reference to estimate the wind speed deficit caused by the wind farm wakes under different wind directions and atmospheric stabilities. For wind farm waked wind directions, the lidar data show that the wind speed is reduced up to 30% at a wind speed of about 10 m/s, depending on atmospheric stability and distance to the wind farm. For illustrating the spatial extent of cluster wakes, an airborne dataset obtained during the scanning wind lidar campaign is used and compared with the mesoscale model with wind farm parameterization and the scanning lidar. A comparison with the results of the model with a wind farm parameterization and the scanning lidar data reveals a relatively good agreement in neutral and unstable conditions (within about 2% for the wind speed), whereas in stable conditions the largest discrepancies between the model and measurements are found. The comparative multi sensor and model approach proves to be an efficient way to analyze the complex flow situation in a modern offshore wind cluster, where phenomena at different length and time scales need to be addressed.*

**Abstract— Error is noted as 30% in the abstract, but 45% in the conclusions. Is this inconsistent or are the authors discussing different points?**

*It was a typo error, now corrected in the paper.*

**page 1 — "which can be transferred to other regions." The meaning of this phrase is not clear.**

*This means that the effect shown in the North Sea could be extrapolated to other offshore regions (e.g. UK, Belgium, ...). To avoid confusion, we have rephrased the sentence in the paper as follows:*

*"In the North Sea, the available offshore area for wind energy is becoming increasingly scarce."*

**page 2 — "the first phase of a potential offshore wind farm to assess accurately the wind resource," preconstruction or resource assessment phase?**

*We understand that preconstruction or resource assessment phase as the same phase.*

Now rephased as:

*...the first phase of a potential offshore wind farm to assess accurately the wind resource, ....*

**page 6, line 127 — It is not clear what these coordinates imply. Azimuth and elevation? The unobstructed azimuth angles visible to the lidar (i.e., everything not within the sector 20◦ to 160◦ )? If this is the case, how are lidar measurements shown in the figures below obtained for this sector?**

*This refers to azimuth where the laser beams of the lidar are not obstructed by a hard target as the wind turbine towers as shown in Figure 5. We have modified the corresponding text to read: "…for a clear view in the azimuthal range [160°, 20°] over the railing to the west."*

**Figure 8: Linear interpolation across such a large range is sure to introduce uncertainty to the results, particularly near the wind plans. How do these measurements compare to the WRF model results?**

*Yes, you are right. We were aware of that, and we use this method as a qualitative inspection of the spatial distribution of the wake. However, now we plot again the flight legs data without any interpolation (see Figure 8a).*

*In Figure 8b we use just the data as measured by the flight using a leg that was passing by the scanning lidar position. The leg position is now clearly shown in Figure 8a to avoid any confusion.*

**Are the green and yellow dashed lines relevant to the information in Figure 8b?**

*No, therefore we now removed them from the plot.*

**Since those wind farms are downstream of the flights, I'm not sure what they communicate. Also, the spacing of dashed lines indicating wind farm clusters is very different from Figure 8a to 8b. Any idea why this is if they are showing measurements along the same transect? "...the closest flight traversal upstream of the lidar measurement point x = −0.33 km)" Can this be made more clear in the figure**

*Please note that the graph in Fig8a is in degrees in Fig8b in meters. Now the lines are removed in Figure 8b as discussed above.*

*The point x = −0.33 km refers to the distance to the right of the scanning lidar. For clarity, it is now rephased in the paper as: "Figure 8 … (x=-0.33 km, where the negative sign refers to the left of the location of the measurements obtained by scanning lidar)"*

**? I believe that this refers to the line closest tp the black diamond in Figure 8a. Is that correct?**

*Yes, for clarity we modify the legend text in Figure 8b.*

**page 14, end of Section 3.1 — How are the different scales reconciled. WRF, Lidar, and the instrumentation on the aircraft all make measurements at significantly different scales and resolutions—effectively observing different phenomena.**

*We make use of collocated measurements, with lidar providing continuous measurements in time at a fix location, and aircraft providing temporal snapshots of the horizontal extent of wakes. Therefore, both data sets are used complementary. The simulations provide both spatial and temporal coverage, but with much coarser temporal and spatial resolution.*

*The lidar, aircraft and WRF results represent a comparison of point, line and three-dimensional spatial data. A qualitative comparison between the WRF and aircraft data should lead one to conclude that the important spatial features of cluster wake regions are well reproduced by the mesoscale model. The direct*

*quantitative comparison between the flight data and point (lidar) measurements place the lidar measurements in the wider context of the cluster flow.*

**Figure 9 — Showing scanning lidar returns would help to understand how the measurements and model results compare. Please show selected scans, or time averages, and give some indication of how much data are collected, measurement uncertainty, etc.**

*Figure 9 reveals the time—space variation in WRF results over the period of the flight measurements. An average over the different panels here is qualitatively representative of the results in Figure 8a.*

*This figure only aims to show what WRF simulations look like for the same date and time period of the measurement transect of the flight shown in Figure 8a and Figure 8b.*

*Now we added the following sentence to the text to help understand how we post-process LOS data to 10-minute averaged data used in the analysis:*

*"In practice, every time a scan is finished, i.e. every 75 s (see key scanning parameters in Table 3), we perform a VAD and store wind speed and wind direction at all five heights. Finally, these data are averaged over 10 min."*

*Number of data (averaged over 10 min) are indicated in Figure13. The measurement uncertainties are shown in* Appendix A.

**Figure 10 — Please show histograms that data are fit to and fit quality. Also, R2 contains fewer observations than the other sectors. How does this impact the fit quality or the analysis?**

*Relative frequency distributions for each sector are shown in Figure 10b. While it is true that there are fewer observations in R2, the number of observations shown in Figure 13 allow an overall statement to be made.*

*As the sectors and the wind farms inside them has not the same density of wind turbines it is not possible to perform a comparison between different sectors. We stated it clearly in the text as "This illustrates that meteorological conditions and wind speed distributions within a region are very different, so that a direct comparison of wind data between the different sectors does not make sense due to the different flow conditions found in each sector."*

**Figure 11 — Unstable conditions look to be bimodal, from WNW and WSW. Both peaks are captured in R4, indicating that there may be problems with statistical convergence in teh other sectors. Observations during neutral conditions are spread over 2 sectors. Does this impact the quality of the analysis?**

*At this particular position in the North Sea, stable conditions predominantly arise from winds from the east, neutral conditions from south-westerly winds, and unstable for either westerly or north-westerly. The bimodal-like appearance of unstable distribution is likely a result of this junction. The main consequence of this directional dependence is the lack of data in the north-west sector for stable conditions.*

**page 18 — "Nevertheless, the reduction in wind speed caused by the wind farms in the other regions can be clearly seen." The polar plots in Figure 12 are very interesting (I love the maps of the wind**

**farms in the background) but are difficult to interpret. Were the WRF simulations really long enough to provide an estimate of wind speed for all the stability conditions and wind direction sectors? Do the lines represent mean wind speeds? Are these the wind speeds at the lidar location or where the lidar scans at 120 m? At what elevation angle? The influence of wind plant wakes is also less pronounced than indicated in the text. Other than perhaps in the stable case (and only for R3), wind plant wakes are difficult to see in Figure 12.**

*The number of data points for each stability and sector are presented above each corresponding bar in Figure 13. The length of the measurement campaign (6 months) constrains the amount of data available in this assessment. However, a few hundred points in each sector for each stability give a representative sampling of the flow conditions. The polar plots indeed represent the mean values at each directional bin. The inclusion of standard deviation was considered but we decided against this since this leads to noisy images.*

*The data are reconstructed from lidar scans at the heights of 40, 80, 120, 160, and 200 m above sea level at a particular point at a distance of 1.5 km West of the lidar itself. The lidar is able to generate a "virtual met mast" over the sea at the point of interest. This in fact represents the novelty of the work. In principle, the lidar could be deployed anywhere within the maximum scanning range (10 km). From Table 3, the elevation at 120 m is 3.70°.*

*Little wake effects are detected in Figure 12 in unstable conditions. More evident are the wakes in neutral conditions for westerly and south-easterly flows. In addition to the sector R3, wakes in stable conditions are also evident in the sector R1, and weaker evidence in the sector R4 where less data are available.*

**Equation (2) — It is unclear whether this metric quantifies error in a time series of velocity (i.e., how closely does WRF match the observed wind speed as a function of time), error between average velocities, stability-binned observations, etc. Please describe the error metric in more detail. Also, this definition casts the WRF simulation as the reference signal. How is uncertainty in the WRF simulations included in the analysis?**

*This quantifies the average error in each wind direction bin for each stability and so we have modified the text to help express this as follows:*

*"… in Figure 13(a) which is computed as an average over all points in each wind direction region for unstable, neutral and stable conditions.*

*In the definition, WRF is used as the reference, as it represents the free wind flow. The decrease of wind speed caused by wakes characterizes the wind speed deficit.*

*"The WRF model setup that was used in this study is based on the extensive set of sensitivity studies carried out in the framework of the NEWA project (Hahmann et al., 2020; Dörenkämper et al., 2020). The final set-up was validated against almost 300 masts in different terrain complexity. In low terrain complexity this set-up showed a bias of the mean wind speed of 0.06 m/s +/- 0.49 m/s evaluated at 110 masts."*

**page 18 — The two error metrics might be more easily understood as εWRF and εWRF-WF.**

As recommended by the reviewer, we changed in the text and in Figure13 udiff(WRF) by $\varepsilon_{WRF}$ und udiff(WRF-WF) by $\varepsilon_{WRF-WF}$.

*The WRF signal is used a reference for the freestream wind speed in the absence of wind farms.*

**Figure 13 — It is difficult to understand what is important in this figure. Why does WRF-WF under or over predict the wind speed in certain sectors and heights? How is the reader to understand where the measurements are made (e.g., at the lidar location, at XX distance along the line of sight, etc.)? In the caption, please note as (a) and (b). Each subfigure has a top row.**

*We see that this could be confusing because of the different vertical scales employed in (a) and (b). This has been corrected in the revised version. For the free wind sectors, little improvement is evident when switching from the WRF with and without wind farm parameterization compared with the waked sectors where an overall reduction in wind speed difference is detected. Essentially, compare the bars in (a) with (b). We do not expect absolute $\varepsilon_{WRF-WF}$ = 0 but rather an improvement.*

*Scan measurement shown are collected at 1.5 km from the system, referred in the text as measurement location. WRF data shown are located at this position as well. Now it is indicated in the text clearer.*

*Results are generally improved when comparing the the wind farm model with wind park parameterization to the model without taking wind parks into account. This reflects the ability of the model to represent wind farm wakes, as they were measured during the same time period. This is to be seen in the tendency for ε to go towards zero when comparing (a) with (b).*

*The caption has been corrected to read (a) and (b).*

**page 21 — Please include a discussion section. It is difficult to tell from the text which results are important, and what the authors are trying to demonstrate. What is the importance of the study?**

*Now a discussion section is included before the conclusion.*

**page 22 — "The smallest, but still most significant differences..." What makes them most significant? Statistically significant? Is significance tested?**

*Now this sentence is deleted.*

**page 23, line 431 — This is unclear, do the authors mean, "... in this way we can reduce, but not eliminate, measurement error."?**

*Correct.*

---

## Author Comment (AC2)

**Manuscript WES-2021-159**

**File: wes-2021-159-RC2-supplement.pdf**

*We thank the reviewer RC2 for reading the manuscript and for the constructive comments and suggestions for improvements. The original reviewer's comments are in black bold, and our response is in black italic font.*

*Beside the current file for the answers to the reviewer, an additional pdf file is attached which refers to the new pdf version of the paper after considering the reviewer's comments. Our changes to the manuscript are additionally presented in blue.*

**General comment**

**I commend the great work done by the authors, especially for the collection of lidar and airborne data. In my opinion, the title of this manuscript is misleading, indeed the reader is going to expect a detailed analysis of state-of-the-art lidar scans to investigate the wind field between wind turbine arrays, while the main focus of the work is an intercomparison between lidar and WRF wind speed at hub height for different wind directions and atmospheric stability regimes, which of course has a more limited scientific interest. Therefore, I am a bit confused about the main insight that this manuscript would convey. At this stage, this manuscript reads more as a technical report rather than a scientific paper. The novelty of this work, if any, should be better emphasized in the manuscript, and discussed in detail.**

*We agree that the focus can be expressed more clearly and could give the reader a wrong idea. Now, we changed the title to:* "Offshore wind farm cluster wakes as observed by a long-range scanning wind lidar measurements and mesoscale modeling"

*We adjusted the abstract to make clear that the focus of the paper. We still think that the manuscript is very useful for the deployment of this relative novel technique in offshore wind energy. Now we included a discussion section where we discuss the novelty of this study.*

1. **L 110-115 - How the various atmospheric stability classes are defined based on the lapse rate? Please provide references as well.**

*There is no clear agreement in the literature on the limit of lapse rate values around neutrality, and for the offshore environment it is even more difficult. We decided to use ±0.04 as a possible limit to filter the data into different stability classes. We have included a reference that gives an overview of this issue and uses a similar limit to ours.*

2. **Fig. 8a - I am not sure it makes sense to generate a color map from the linear interpolation of the data collected over the transects. I think it would more informative to show the map with the transect locations and overlap the wind data with linear plots.**

*Yes, you are right. We were aware of that, and we use this method as a qualitative inspection of the spatial distribution of the wake. However, now we plot again the flight legs data without any interpolation (see Figure 8a).*

*In Figure 8b we use just the data as measured by the flight using a leg that was passing by the scanning lidar position. The leg position is now clearly shown in Figure 8a to avoid any confusion.*

3. **Fig. 10 b - I would plot the experimental probability density functions of the wind data, then overlap the respective Weibull distribution and/or the Weibull factors, as reported in the legend.**

*Thank you for your suggestion. Now a plot with the data and their fit can be found in the appendix B to avoid a large number of graphs in the main text.*

4. **L 318 – "The strong fluctuations in the wind speed of the lidar data in region R1 are due to the very small amount of data in neutral stratification" Can you try to quantify the accuracy of the data through any statistical approach, e.g. error on the mean or percentiles obtained through bootstrapping?**

*Here we are not saying that the lidar data is in error, but rather that we do not have enough data. For a quantification of uncertainties in the lidar data, please see the appendix A.*

---

## Author Response (AR2)

Manuscript WES-2021-159

*We would like to thank the reviewers and the editor for their efforts in checking again the manuscript. Below, the reviewer's comments are in bold black and our direct responses to the comments in blue italics. The two points below have been implemented in the revised manuscript.*

**Comments to the author**:
**Dear Authors,**

**Both reviewers consider your revisions sufficient for your manuscript to be accepted for publication in WES. However, before I can accept your work for final publications, I have two additional requests.**

**1. please correct your title: either "by a long range scanning wind lidar and ..." or "by long range scanning wind lidar measurments and ..."**
*Offshore wind farm cluster wakes as observed by long range scanning wind lidar measurements and mesoscale modeling*

**2. The reviewers in particular appreciated the valuable data set that you generated. I appreciated from your Data Availability statement that you are not committing to sharing your data open source at this time. However, as I minimum requirement, I would like you to share the final processed data (incl. possibly the script files) that were used to generate the figures in your manuscript. Please update your statement accordingly, and provide the data (e.g. on figshare, zenodo, ...). [There should be no legal or other issues with this, since, as you know, many tools exist online to extract these data from your figures anyway]**

*Data of the key paper figures have been uploaded to*
*https://doi.org/10.6084/m9.figshare.19747252.v1*

**Sincerely,**
**Johan Meyers**

---

## Author Response (AR3)

Manuscript WES-2021-159

*We would like to thank the reviewers and the editor for their efforts in checking again the manuscript. Below, the reviewer's comments are in bold black and our direct responses to the comments in blue italics.*

**22 May 2022**

**Associate Editor decision: Publish subject to minor revisions (review by editor)**

**By Johan Meyers**

**Comments to the author:**
**Dear Authors,**

**Thank you for publishing the key figures to the figshare. However, please, simply provide thee data of all figures in the figshare.**

**Sincerely,**
**Johan Meyers**

*Dear Editor,*

*All data relevant to the modeling have been made available on figshare. The information contained in the remaining figures is for conceptual purposes to support the scientific work presented in the main text and, in our view, not relevant for such applications. We are not aware from previous WES manuscripts that we have to disclose the data of all figures.*

*As mentioned in the manuscript, all data from the X-Wakes project will be made available upon request once the project is completed and the reader can contact us.*

*Yours sincerely,*

*The Authors*